# Intriguing Properties of Randomly Weighted Networks: Generalizing while Learning Next to Nothing

**Amir Rosenfeld, John K. Tsotsos**
Department of Electrical Engineering and Computer Science
York University, Toronto, ON, Canada
`amir@eecs.yorku.ca,tsotsos@cse.yorku.ca`

## Abstract

Training deep neural networks results in strong learned representations that show good generalization capabilities. In most cases, training involves iterative modification of all weights inside the network via back-propagation. In this paper, we propose to take an extreme approach and fix *almost all weights* of a deep convolutional neural network in their randomly initialized values, allowing only a small portion to be learned. As our experiments show, this often results in performance which is on par with the performance of learning all weights. The implications of this intriguing property or deep neural networks are discussed and we suggest ways to harness it to create more robust representations.

## 1 Introduction

Deep neural networks create powerful representations by successively transforming their inputs via multiple layers of computation. Much of their expressive power is attributed to their depth; theory shows that the complexity of the computed function grows exponentially with the depth of the net (Raghu et al. (2016)). This renders deep networks more *expressive* than their shallower counterparts with the same number of parameters. Moreover, the data representation is more efficient from an information-theoretic point of view (Shwartz-Ziv & Tishby (2017)). This has led to increasingly deeper network designs, some over a thousand layers deep (He et al. (2016)).

Current optimization methods (e.g., SGD) update all weights of the network to minimize some loss function. Could be it that not all weights need updating, or are equally useful? Modern day architectures (Krizhevsky et al. (2012); Simonyan & Zisserman (2014); He et al. (2016); Zagoruyko & Komodakis (2016)) contain millions to billions of parameters (Shazeer et al. (2017)) - often exceeding the number of training samples (typically ranging from tens of thousands (Krizhevsky & Hinton (2009)) to millions (Russakovsky et al. (2015))). This suggests that these networks could be prone to over-fitting, or are otherwise highly-overparameterized and could be much more compact. It also may explain why current methods in machine learning tend to be so data-hungry.

Instead of training all network weights, we suggest the almost extreme opposite: network weights are initialized randomly and only a certain fraction are updated by the optimization process. While this has a negative effect on network performance, the magnitude of this effect is surprisingly small with respect to the number of parameters not learned.

This effect holds for a range of architectures, conditions, and datasets. To the best of our knowledge, while others have analytically analyzed properties of randomly weighted networks, we are the first to explore the effects of keeping most of the weights at their randomly initialized values in multiple layers. We claim that successfully training mostly-random networks has several implications for the current understanding of deep learning, specifically: (1) Popular network architectures are grossly over-parameterized, and (2) Current attempts at interpreting emergent representations inside neural networks may be less meaningful than thought.

## 2 RELATED WORK

**Random Features**: there is a long line of research revolving around the use of random features in machine learning. Extreme Learning show the utility of keeping some layer of a neural net fixed - but this is usually done only for one or two layers, and not within layers (Huang et al. (2015)). Rudi & Rosasco (2017) has shown how picking random features has merits over matching kernels to the data. have analytically shown useful properties of random nets with Gaussian weights Giryes et al. (2015). **Net Compression/Filter Pruning**: many works attempt to compress the net after learning (Li et al. (2016); Han et al. (2015)). **Network Interpretability**: recent works, such as that of Bau et al. (2017) have attempted to show how emergent representations relate to "interpretable" concepts, and some have tried to supervise the nets to become more interpretable (Dong et al. (2017)). As we keep most features random, we argue that (nearly) similarly powerful features can emerge without necessarily being interpretable.

## 3 METHOD & EXPERIMENTS

We begin with some definitions: let $\mathcal{W}$ be the set of all parameters of a network $N$. All of our experiments can be framed as splitting $\mathcal{W}$ into two disjoint sets: $\mathcal{W} = w_f \cup w_l$. In each experiment we fix the weights of $w_f$ and allow $w_l$ to be updated by the optimizer. $w_f$ are either randomly initialized or set to zero. Let $F = \{F_1 \dots F_n\}$ be the filters of the conv. layers of $N$ (including shortcut-layers for res-nets). $w_l$ defines a subset $f_i \subseteq F_i$ for each $i \in \{1 \dots n\}$. We test how well the network converges to a solution for various configurations of $w_l$.

EXPERIMENTS

We experiment with the CIFAR-10 and CIFAR-100 datasets (Krizhevsky & Hinton (2009)) and several architectures: wide-residual networks, densely connected resnets, AlexNet, and VGG-19 (resp. Zagoruyko & Komodakis (2016); Huang et al. (2016); Krizhevsky et al. (2012); Simonyan & Zisserman (2014)). For baselines, we modify a reference implementation [1]. To evaluate many different configurations, we run most of our experiments for 10 epochs, with the default hyper-parameters. For a few experiments we perform a full run. We test the following configurations:

**Fractional Layers:** Setting the fraction $p_i = \frac{f_i}{F_i}$ of filters of all layers $L$ of $N$, except the fully-connected (classification) layer. We do this for the following fractions: $.07, .08, .09, .1, .4, .7$ (less than $0.07$ will mean no filters for networks where $F_0 = 16$ e.g., in wide-resnets). **Integer number of filters:** learning a fixed integer $k_i \in \{1, 5, 10\}$ of filters per layer. We show that learning a *single filter in each layer* leads to non-trivial performance for some architectures. **Single Layers:** freezing all weights except those of a single block of layers. This is only for the wide and dense resnets. The blocks are selected as $w_l \in \{conv_1 block_1, block_2, block_3, fc\}$ where $conv_1$ is the first convolutional layer, $block_i$ is one of 3 blocks of a wide-resnet with 28 layers & widen factor of 4 or of a densenet with a depth of 100 and a growth-rate of 12. Parameters of all BN layers are always learned.

Fig. 1 shows the top-1 accuracy after 10 training epochs. Shaded areas specify integer number of filters. The best performer is densenets (orange). AlexNet (blue) failed to learn for non-trivial fractions or for only a few filters per layer. Note the gap between the fixed weights and those zeroed-out (faded). Zeroing out the weights effectively reduces the number of filters from the network. Using 70% of filters while zeroing the rest out achieves the same performance for densenets. Interestingly, learning only a *single filter per layer can* result in a non-trivial accuracy. In fact, zeroing out all non-learned weight, resulting in a net with a single-filter per layer, still is able to do much above chance, around 45% on CIFAR-10 with resnets.

This is better seen in Fig 2 which plots the performance obtained vs. the total number of parameters on a logarithmic scale. Even when zeroing out all non-learned parameters, wide-resnet obtains decent performance with less that 100K parameters.

Finally, we learn only subsets of layers out of the layers specified and report the best performance for each block in Table 1 (b). Notably, for cifar-100 it proved much better to learn an internal block's parameters than those of the fully-connected layer.

---

[1] https://github.com/bearpaw/pytorch-classification

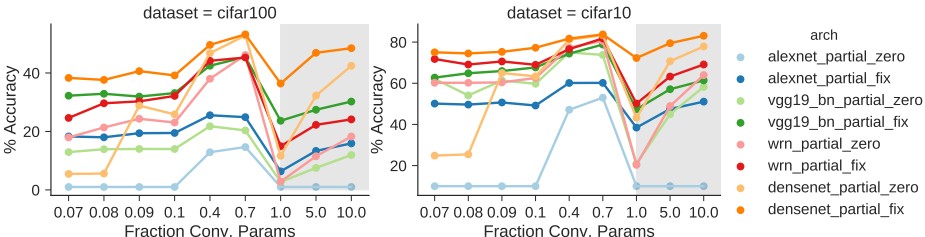

Figure 1: Training only a few parameters: deep networks can generalize surprisingly well when only a small fraction of their parameters is learned. Unshaded area specifies fraction of filters learned for at each conv. layer. Shaded area specifies an integer number of filters learned. Faded lines are for performance where all weights are set to 0 except the learned fraction.

| Method | Fraction | No. Learned Params $\times 10^6$ | Perf | Perf† |
|---|---|---|---|---|
| wide-resnets | 0.1 | 3.66 | 94.12 | 91.53 |
| wide-resnets | 0.4 | 14.6 | 95.75 | 95.49 |
| densenets | 0.1 | 0.09 | 88.73 | 82.11 |
| densenets | 0.4 | 0.3 | 93.33 | 92.46 |

(a)

| Block | Perf (C10/C100) |
|---|---|
| conv1 | 64.7/15.1 |
| block1 | 73/22.9 |
| block2 | 76.3/28 |
| block3 | 76.3/32 |
| fc | 68/33 |

(b)

Table 1: (a) Performance vs fraction of parameters learned on CIFAR-10 (full training, 200/300 epochs). † means performance when $w_f$ (fixed parameters) are *all set to zero*. (b) learning only a single block (only 10 epochs). A gray line signifies densenet got better performance for this block on both datasets.

**Full Runs** We also ran full training sessions for wide-resnet and densenet (200,300 epochs respectively) with a limited no. of parameters on CIFAR-10. The results are summarized in Table 1. Specifically, wide resnets, when 60% of the filters are arbitrarily zeroed out, achieve almost the baseline performance of 96.2%. Please refer to Table 1 (a).

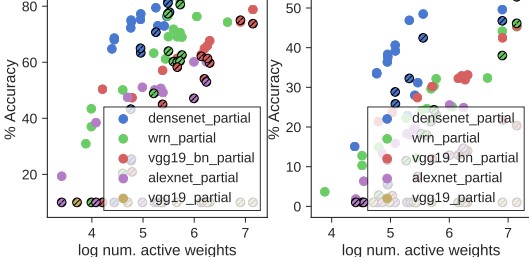

Figure 2: Performance vs. absolute number of parameters (log scale). Much of the performance can be preserved by learning a relatively smaller no. of parameters, and even zeroing out the rest (thatched circles). densenet (blue) is very efficient in this sense.

## 4 DISCUSSION

We have demonstrated that learning only a small subset of the parameters of the network or a subset of the layers leads to an unexpectedly small decrease in performance (w.r.t full learning) - even though the remaining parameters are either fixed or zeroed out. This is contrary to common practice of training all network weights. We hypothesize this shows how over-parameterized current models are, even those with a relatively small number of parameters, such as densenets. Three simple applications of this phenomena are (1) cheap ensemble models, all with the same "backbone" fixed network, (2) learning multiple representations with a small number of parameters added to each new task and (3) transfer-learning by learning a *middle* layer vs the final classification layer.

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
