# OpenReview forum: "Intriguing Properties of Randomly Weighted Networks: Generalizing while Learning Next to Nothing"
_ICLR.cc/2018/Workshop — Reject_

### Official Review · AnonReviewer2 · 2018-03-04
**Interesting results but not surprissing**

**Rating:** 4
**Confidence:** 5

**Review:**

This paper analyzes parameter redundancy in current large networks and how networks using lower number of parameters can still do a decent job in generalization.

While this is interesting, it is not new. It is well known that redundancy exists and that is why compression among other (post-processing) techniques exist. The paper also suggests some potential ways to make use of the extra computational capacity, however, there are only ideas and no results. Would be nice to see these ideas in practice. Redundancy is mostly needed for helping the optimizer, while reducing the number of parameters helps in some tasks there is no proof that it works in larger datasets (ImageNet or larger).

In summary, this is a nice analysis of redundancy in neural networks in small datasets but there is not strong take home message.

---

### Official Review · AnonReviewer3 · 2018-03-09
**Interesting work**

**Rating:** 7
**Confidence:** 4

**Review:**

The authors train several network architectures while fixing parts of the parameters to the initalization. They show that even when fixing a large subset of the trainable parameters, the networks perform reasonably (but worse than when training parameters).

The work seems solid but could be improved by a few comparisons. Especially I'm wondering how much the fixed parameters contribute to the network performance. The zeroing-out-experiments explore this question but I would be interested to see how good the models perform when zeroing out the fixed parameters already while training (of course this doesn't work when fixing a full layer). This could serve as a simple baseline with the same number of parameters. For example, I'm not sure how impressive the performance on CIFAR100 is when fixing more than 60% of the weights. Also I'm missing the performance of the fully trained models (e.g. is a highly reduced densenet still better than a fully trained AlexNet?).

The authors explore a very interesting idea and I think especially the implications of interpreting representations are highly relevant.

Overall I think this work is worth presenting at the ICLR workshops.

---

### Official Review · AnonReviewer1 · 2018-03-10
**final review**

**Rating:** 5
**Confidence:** 3

**Review:**

Learning a fraction of filter weights while with other weights not updated is an interesting setting. The paper shows that such partially trained network is still able to produce reasonably well results. The authors also noted that zeroing the freezed weights would hurt the performance. I'm curious to see what happens if these un-learned weights are zeroed before training starts. If that still leads to worse results, than it means the distribution for those random weights matters. Probably the paper can further explore when and how the random weights help training even they are not trained.  Overall, I think the results in this paper are still quite preliminary, and not much conclusion can be drawn from them.

---

### Comment · AnonReviewer1 · 2018-02-24
**interesting finding, but incompelete experiements**

This paper shows that deep networks with a large portion of fix weights can still generalize quite well. The oberservation is quite interesting, but not that suprising. As existing work (including some of the references in this paper) has already shown that random networks with only the last layer trained are quite effective. In addition, it would be important to compare with a small network with the same number of trainable parameters.

---

> ### Author Response · Authors · 2018-02-25
> **Reply**
>
> The authors are thankful for the comments.
> A much more detailed version, with more experiments an analysis is now available on arXiv,
> https://arxiv.org/abs/1802.00844
> Regarding previous work: To which work does the reviewer refer, showing that training only the last layer is effective?
> This was show, and to a limited extent, in the context of transfer learning, which is not quite the context here. Indeed, the experiments include attempting to train only the last layer, which results in very poor performance w.r.t many of the explored alternatives.
> Additionally, none of the previous works have tried to train only a subset of features in each layer, let alone try to train only a middle layer, or only the first layer, as the proposed work.
> The exact number of parameters is not a good predictor of network performance, for example densenet vs. wide-residual networks.
> While the experiments did not include an exact version of what is suggested here, training smaller networks with the same number of trainable parameters, we did train on a variety of architectures, varying the number of trainable parameters in each, as well as showing that the untrained parameters are not left unused, as simply zeroing them out significantly hinders the network performance.

---

### Author Response · Authors · 2018-02-25
**Code Available**

An preliminary version of code to reproduce the experiments is now available on GitHub.
https://github.com/rosenfeldamir/pytorch-classification

---

### Decision · Program_Chairs · 2018-03-20
**ICLR 2018 Workshop Acceptance Decision**

**Decision:**

Reject

**Comment:**

Based on the reviews, this paper has not been accepted for presentation at the ICLR workshop. However, the conversation and updates can continue to appear here on OpenReview.